# Transferability of ISSR, SCoT and SSR Markers for *Chrysanthemum* × *Morifolium* Ramat and Genetic Relationships Among Commercial Russian Cultivars

**DOI:** 10.3390/plants10071302

**Published:** 2021-06-27

**Authors:** Lidia S. Samarina, Valentina I. Malyarovskaya, Stefanie Reim, Lyudmila G. Yakushina, Natalia G. Koninskaya, Kristina V. Klemeshova, Ruset M. Shkhalakhova, Alexandra O. Matskiv, Ekaterina S. Shurkina, Tatiana Y. Gabueva, Natalia A. Slepchenko, Alexey V. Ryndin

**Affiliations:** 1Federal Research Centre the Subtropical Scientific Centre of the Russian Academy of Sciences, 354002 Sochi, Russia; lab-bfbr@vniisubtrop.ru (V.I.M.); vishnya584@yandex.ru (L.G.Y.); natakoninskaya@mail.ru (N.G.K.); klemeshova_kv@mail.ru (K.V.K.); shhalahova1995@mail.ru (R.M.S.); matskiv_a@mail.ru (A.O.M.); shurkina-ekaterina@rambler.ru (E.S.S.); tatyana_litus@mail.ru (T.Y.G.); slepchenko@vniisubtrop.ru (N.A.S.); ryndin@vniisubtrop.ru (A.V.R.); 2Institute for Breeding Research on Fruit Crops, Julius Kühn-Institut, Federal Research Centre for Cultivated Plants, 01326 Dresden, Germany; stefanie.reim@julius-kuehn.de

**Keywords:** *Chrysanthemum* × *morifolium*, ornamental crops, genetic diversity, ISSR, SCoT, SSR

## Abstract

Characterization of genetic diversity in germplasm collections requires an efficient set of molecular markers. We assessed the efficiency of 36 new SCoT markers, 10 new ISSR markers, and 5 microsatellites for the characterization of genetic diversity in chrysanthemum core collection of 95 accessions (Russian and foreign cultivars). Seven new SCoT (SCoT12, 20, 21, 23, 29, 31, 34) and six new ISSR markers ((GA)8T, (CT)8G, (CTTCA)3, (GGAGA)3, (TC)8C, (CT)8TG) were efficient for the genetic diversity analysis in *Chrysanthemum* × *morifolium* collection. After STRUCTURE analysis, most Russian cultivars showed 20–50% of genetic admixtures of the foreign cultivars. Neighbor joining analysis based on the combination of SSR, ISSR, and SCoT data showed the best accordance with phenotype and origin compared to the separate analysis by each marker type. The position of the accessions within the phylogenetic tree corresponded with the origin and with some important traits, namely, plant height, stem and peduncle thickness, inflorescence type, composite flower and floret types, flower color, and disc color. In addition, several SCoT markers were suitable to separate the groups distinctly by the phenotypical traits such as plant height (SCoT29, SCoT34), thickness of the stem and peduncle (SCoT31, SCoT34), and leaf size and the floret type (SCoT31). These results provide new findings for the selection of markers associated with important traits in Chrysanthemum for trait-oriented breeding and germplasm characterization.

## 1. Introduction

*Chrysanthemum* × *morifolium* Ramat is an herbaceous, perennial, ornamental plant that belongs to the family of *Compositeae* (*Asteraceae*). After roses, chrysanthemum occupies the second place on the world ornamental plant sales list [1]. There are more than 20,000 chrysanthemum cultivars in the world, of which 90% were developed by conventional breeding techniques [2]. Cultivated chrysanthemum are allo-hexaploids and aneuploids with the most frequent somatic chromosome number of 2n = 6x = 54 [3]. The breeding constrains in chrysanthemum are referred to its genome complexity, high level of heterozygosity, and the occurrence of both inbreeding depression and self-incompatibility [4].

The conventional Chrysanthemum breeding in Russia is mainly aimed to develop spray and pompon type bouquet cultivars which are tolerant to different types of stress. Dozens of locally adapted cultivars have been developed during the over 50-year history of breeding in FRC SSC RAS [5]. This collection includes a broad set of frost tolerant and disease tolerant genotypes. Chrysanthemums are the most popular potted, cut, and garden ornamentals in Russia with the abundant diversity of flower type, colour, and plant architecture. This diversity is the results of germplasm exchange, open pollination, and a high level of heterozygosity. During the recent years, a comprehensive phenotypical characterization of the Russian core collection has been performed; however, until now, the genetic origin, relationships, admixtures, and genetic structure of these germplasm have not been evaluated. 

Application of molecular markers is an efficient tool for germplasm characterization and for trait-oriented breeding of chrysanthemum worldwide. Several recent publications showed the genetic diversity in cultivated chrysanthemums using AFLP, ISSR, SRAP, SCoT, and SSR markers [6,7,8,9,10,11,12]. 

Among the different marker types, co-dominant nuclear microsatellite markers (SSRs) have desirable advantages for assessing the genetic features of species at individual and population levels, such as locus specificity, high reproducibility, technical simplicity, and polymorphism [13,14]. Despite on the absence of the whole genome sequencing, a set of SSR markers was recently developed for Chrysanthemum [4,10]. 

Inter simple sequence repeats (ISSRs) are another efficient marker type that is multilocus, dominant, reproducible, and highly polymorphic for genetic diversity studies [15]. ISSR repeats are believed to be present mostly in the non-coding regions of chromosomes and specific stretch of DNA sequences which are not active [16]. High repeatability could be observed if the ISSR primers have sufficient specificity [17,18]. Some studies also reported on the usefulness of several ISSR markers for chrysanthemum genetic diversity evaluation [7,8]. The high occurrence of ISSR between normal coding genes and their presence in certain chromosomes as satellite bodies makes ISSR unique and advantageous to be used for DNA fingerprinting. 

Start codon targeted (SCoT) markers are based on polymorphism in the short, conserved region in plant genes surrounding the ATG translation initiation codon. It is possible that some SCoT markers would be codominant due to insertion–deletion mutations [19]. Since the region flanking the ATG start codon is highly conserved in all plant species, it was predicted that the SCoT method would be useful for generating DNA markers in diverse plant species [20]. One study was published recently on the efficiency of few SCoT markers for several chrysanthemum cultivars [21].

Each marker type has its advantages and disadvantages; thus, the combination of several marker types can be helpful for better understanding of genetic diversity and structure in the collections. In this study, we evaluated the efficiency of 36 SCoT markers, 10 new ISSR markers, along with the 5 SSRs for the genetic diversity analysis of *Ch. morifolium*. In total, 95 germplasm accessions derived by foreign and Russian breeders were evaluated in this study. Based on the genetic data, we (I) evaluated the efficiency of the new ISSRs and SCoTs for the molecular characterization of this collection; (II) estimated the genetic diversity, genetic structure, and relationships within Russian and foreign chrysanthemum genotypes, and (III) revealed correlations between molecular and phenotypical data. These results are important for germplasm characterization and for breeding of chrysanthemums.

## 2. Results

### 2.1. Transferability and Discriminating Power of SCoT, ISSR, and SSR Markers for Chrysanthemum

Out of 36 SCoT primers, six primers (SCoT7, SCoT8, SCoT9, SCoT10, SCoT26, SCoT27) showed no amplification in chrysanthemum, thirteen primers (SCoT1, SCoT2, SCoT3, SCoT5, SCoT15, SCoT16, SCoT24, SCoT25, SCoT28, SCoT30, SCoT32, SCoT35, SCoT36) showed low quality amplification quality with weak or fuzzy bands, and ten primers (SCoT4, SCoT6, SCoT11, Scot13, SCoT14, SCoT17, SCoT18, SCoT19, SCoT21, SCoT22) showed identical amplification patterns with no polymorphism. These 29 markers were removed from the analysis. The remaining seven SCoT primers showed reproducible results with clear polymorphisms and resolution within chrysanthemum genotypes (SCoT12, SCoT20, SCoT23, SCoT29, SCoT31, SCoT33, SCoT34) confirming their transferability to chrysanthemum (Appendix A). 

With the seven SCoTs, a total of 71 bands were detected and ranged from 4 (for SCoT23) to 15 (for SCoT12) (Table 1). The average polymorphism (P) in the chrysanthemum collection was 54.5%, ranging from 45.7 (for SCoT29) to 73.8 (for SCoT12). With the seven SCoTs, an average polymorphism information content (PIC) of 0.39 was detected with the highest value of 0.42 for SCoT12. The mean discriminating power (D) was 0.79% and the mean diversity index was *h* = 0.49 with 10% of monomorphic bands observed. 

Out of 10 ISSR primers, two primers (ISSR851; ISSR14) showed no amplification and two primers showed low quality amplification (ISSR813; ISSR13) in chrysanthemum. These four primers were removed from the analysis. The six remaining ISSRs showed reproducible results with clear polymorphisms and resolution within chrysanthemum genotypes (ISSR810, ISSR815, ISSR873, ISSR880, ISSR15, ISSR814.1). 

Using these six ISSRs, a total of 118 bands were detected that ranged from 5 (for ISSR13) to 24 (for ISSR873) (Table 1). The average P was 69.0% and ranged from 48.7 (for ISSR810) to 92.3 (for ISSR873). An average PIC of 0.38 was detected with the highest value of 0.44 for ISSR873. The mean discriminating power (D) was 0.89% and the mean diversity was h = 0.43 with 2% of monomorphic bands observed. Five SSR markers showed comparatively weak polymorphisms within the chrysanthemum germplasm. A total of 15 bands were detected. The average P was 28.2% and ranged from 12.8 (for SSR357) to 50.3 (for SSR320). An average PIC of 0.45 was detected with the highest value of 0.50 for SSRgi298296818. The mean discriminating power (D) was 0.41%, the mean diversity was h = 0.31 for the selected SSR markers, and 25% of monomorphic bands were observed. 

Identical DNA-fingerprints were not observed using SCoT and ISSR markers; however, after SSR analysis, 36 accessions out of 95 chrysanthemum accessions showed identical genetic patterns. 

### 2.2. Genetic Structureof Germplasm Collection Based on SSR, ISSR, and SCoT Polymorphisms

During STRUCTURE analysis, the number of genetic clusters was estimated separately for SSR, ISSR, and SCoT genetic data. Following STRUCTURE HARVESTER analysis, SSR, SCoT, and ISSR markers showed K = 2, K= 6, and K = 3 genetic cluster, respectively. 

According to SSR data, the 95 chrysanthemum accessions were divided only in two genetic groups. The first group (red color) combined 24 accessions of local and foreign origin, but most of them with spray inflorescence and compound corymbs. The second big group (green color) consisted of the remaining 72 accessions of different origin with the different phenotypical traits (Figure 1top).

SCoT data did not result in a clear genetic structure and most of the accessions showed similar patterns of distribution of amplified fragment (Figure 1middle). However, 19 accessions had high percentage of genetic admixtures of genetic cluster 1 (blue color), namely ‘Barca’, ‘Anastasia Green’, ‘Anastasia Star Pink’, ‘Ariana Lime’, ‘Baltica White’, ‘Wilhelmina’, ‘Gagarin’, ‘Grand Pink’, ‘Regina’, ‘Regina white’, ‘Sevan’, ‘Statesman’, ‘Annecy White’, ‘Magnum’, ‘Jaguar Purple’, ‘Vitchizhna’, ‘Dodu’, ‘PIP’, and ‘PIP Salmon’. These accessions shared common phenotypical traits such as thick peduncle, pompon type, and grey shadow in the leaf color. 

According to the ISSR data, 95 accessions were grouped into three genetic clusters (K = 3) (Figure 1bottom); however, many accessions showed significant admixtures of the other clusters. The cluster 1 (red) combined 29 accessions, of which 17 had remarkable genetic admixtures of 20–50% of the clusters 2 and 3. This cluster mostly contained foreign accessions of a different phenotype. Only two Russian cultivars (‘Zolotaya Osen’ and ‘Noktyurn’) joined this cluster. The cluster 2 (green) combined 27 accessions, and eight of them had remarkable genetic admixture of 20–50% of the clusters 1 and 3. This cluster mostly contained Russian hybrids and cultivars with similar phenotypical traits, e.g., small composite flowers. The most accessions in this cluster were developed in Russian breeding programs from the common ancestors. However, seven foreign genotypes which were used as parents in the local breeding programs joined this cluster, namely ‘Desna’, ‘Princess Anna’, ‘Regina’, ‘Etrusko’, ‘Jaguar purple’, ‘Mona Lisa’, and ‘Ksenia’.

The cluster 3 (blue) combined 39 accessions, and 26 of them had remarkable genetic admixtures of 20–50% of the cluster 1 and 2. This cluster contained mostly the new modern foreign cultivars with pompon inflorescence type. Seven Russian genotypes also joined this cluster, namely ‘Goryanka’, ‘Medeya’, and ‘Simfoniya’ along with four hybrids.

Joint STRUCTURE barplot based on the three combined marker types was not informative enough to detect a clear genetic structure of the collection.

### 2.3. Genetic Diversity and Relationships and Correspondence with Phenotypical Traits 

The following neighbour joining analysis of 88 accessions (the accessions without missing data) was performed based on the combined SSR, SCoT, and ISSR data as well as separately for each single marker type. Additionally, a neighbour joining analysis was performed based on 20 flower traits (the period of flowering, the plant height, and flower characteristics). The phylogenetic tree based on the combination of all marker types (Figure 2) showed the best accordance to the phenotypic tree (Figure 3). Therefore, the combined genetic tree was selected for the subsequent interpretation of the genetic relationship within the core collection. Several distant branches were observed in the joint genetic tree (Figure 2).

The branch I included six accessions of foreign origin (Figure 2 and Appendix A). All these cultivars belonged to the phenotypic cluster II (Figure 3) which have common traits such as big composite flowers and strong growth (Table 2). 

The branch II combined 13 accessions that were separated in two groups: the first group included seven spray genotypes with three locally bred hybrids; the second small group combined foreign cultivars with big pompon type, namely ‘Princess Armgard Bronze’, ‘Saffina’, ‘Spider Pink’, ‘Rebonnet’, and ‘Angelys Jaune’. 

The branch III was the most abundant and combined the six sub-branches. The branch IIIa combined all disbud and pompon foreign cultivars such as ‘Desna’, ‘Cassandra’, and ‘Saratov’. Interestingly, the position of the cultivars corresponded with their position in the phenotypic tree—cluster II (Figure 3; Table 2). The branch IIIb included 12 accessions with small composite flowers in spray inflorescence. Several locally derived hybrids (Р-194-13, Р-195-7, Р-195-8, Р-195-9, P-194-12) joined this group. Their position was in accordance with the phenotypic tree where all these accessions belong to the cluster I (Figure 3). ‘Izetka Bernstein’, ‘Baltica’, and ‘Vesuvio’, which are often used in the Russian breeding programs, also joined this group.

The biggest sub-branch IIIc was divided into four sub-sub-branches: IIIc1 (Figure 2) included mostly the spray cultivars belonged to phenotypic cluster II (Figure 3); IIIc2 included 14 accessions of different flower type which are mostly foreign disbud cultivars and belonged to phenotypic cluster II, except the local cultivar ‘Zolotaya Osen’; IIIc3 represented a big branch of 31 accessions divided into the two main groups. Most Russian genotypes were combined this IIIc3 group. One sub-group of IIIc3 combined 14 assessions mostly with small spray and anemone-shaped composite flower such as ‘Statesman’, ‘Rozovaya Dragocennost’, ‘Nejnost’, ‘Krasnoe Znamya’, ‘Medeya’, and ‘Focus’. All these accessions belonged to phenotypic cluster I (Figure 3). The next sub-group of IIIc3 consisted of 16 accessions representing a mixed group with mostly spray (‘Sadko’, Р-196-15, Р-196-26, Р-192-12, Н-103-7) or disbud big flowered inflorescence, such as ‘Etrusko’, ‘Mirazh’, ‘Rezume’, and ‘Regina’. Interestingly, the cultivar ‘Rossano’ was the only accession placed separately in sub-branch IIIc4. 

To conclude, the positions of accessions in the phenotypic tree were generally consistent with the genetic tree. However, the genetic data provided a better resolution and cluster separation of the core collection according to origin and specific traits, especially the composite flower size. 

Among the other phenotypical traits, the following traits were distinguishable by genetic branches: plant height, stem thickness, peduncle thickness, inflorescence type, composite flower type, floret type, and disc colour (Table 2).

PCA was performed separately for the SCoT markers in order to check the best correlations with phenotypic traits. Some SCoT markers, namely SCoT20, SCoT23, and SCoT34, showed a clear separation of groups by phenotypical traits of chrysanthemums. Each polymorphic fragment was verified and some of the fragments were trait-specific. The genotypes belonged to the certain cluster shared common traits. For example, three big distant groups were observed by SCoT20 data (Figure 4). The group I mostly consisted of the Russian bouquet cultivars and hybrids with medium stem thickness: ‘Goryanka’, ‘Nezhnost’, ‘Rozovaya Dragocennost’, ‘Sadko’, and ‘Krasnoe Znamya’ et al. The group II consisted of foreign bouquet cultivars with thick stem such as ‘Resume’, ‘Grand Pink’, ‘Cassandra’, ‘Anastasia Green’, ‘Balloon’, ‘Bigoudi Purple’, and ‘Bigoudi Red’ et al. Finally, the group III consisted of disbud cultivars with white pompon flowers with thick peduncle and big leaves, such as ‘Baltica White’, ‘Gagarin’, ‘Zembla White’, ‘Wilhelmina’, and ‘Regina’. 

Similarly, marker SCoT23 revealed four distant groups. The groups I and II combined mostly tall plants of 120–160 cm height. On the other hand, groups III and IV combined short plants up to 100-cm height with doubled composite flowers. In addition, a big capitulum diameter was observed in all cultivars that belonged to the group I, but in the group II, it was of medium size (Figure 4; Appendix A). In addition, some other markers separated the groups distinct by the phenotypical traits such as plant height (SCoT29, SCoT34), thickness of the stem and peduncle (SCoT31, SCoT34), and leaf size and the floret type (SCoT31) (data are not illustrated). 

## 3. Discussion

### 3.1. Transferability and Discriminating Power of SCoT, ISSR, and SSR Markers for Chrysanthemum

Chrysanthemums are mostly self-incompatible ornamental plants with a highly heterozygous genome, and most cultivars are hexaploids or aneuploids [3,21,22]. Because of genome complexity, the marker assistant breeding of chrysanthemum is challenging and still needs efficient sets of molecular markers. This study reports the efficiency of several SSR, SCoT, and ISSR markers for the analysis of genetic diversity and for establishing genetic relationships among popular Russian cultivars in comparison with the famous foreign cultivars. It is well known that multilocus DNA markers (such as ISSRs and SCoTs) are often transferable to different plant species and genera; however, the efficiency of their application can vary significantly depending on plant species [15,19,20]. 

SCoT markers have been successfully applied for diversity analysis and finger-printing in chrysanthemum [9,11] and many other crops. The detection is agarose gel-based and, therefore, simple and relatively cheap to use. Compared with the previous studies on chrysanthemum which evaluated seven SCoT markers [9] and eight SCoT markers [11], in this study, an increased set of 36 SCoT markers was evaluated. However, the main part of the SCoT marker tested in our study was not polymorphic and six markers were not amplified. Only seven of them were efficiently amplified and polymorphic for *Ch. morifolium* germplasm. Although some markers differed in only one nucleotide (SCoT12 and SCoT13; SCoT29 and SCoT30), they produced very different DNA marker profiles. This is consistent with a previous study on roses [19]. 

A few ISSRs were recently reported to be efficient for chrysanthemums; thus, in this study, we selected several ISSRs which are placed between the following genome regions in chrysanthemum: (GA)8T, (CT)8G, (CTTCA)3, (GGAGA)3, (TC)8C, (CT)8TG. ISSR813 and ISSR815 primers that differed only at one anchored nucleotide showed different efficiency in chrysanthemum. Interestingly, tetranucleotide repeats ISSR873 and ISSR880 showed a good transferability for *Chrysanthemum × morifolium* despite the fact that tri- and tetra-nucleotides are less frequent and their use is less than for di-nucleotides ISSRs [15] In addition, dinucleotide repeats CT and TC were polymorphic and reproducible in chrysanthemum when anchored with G (ISSR815), C (ISSR15), and TG (ISSR814.1). This is in accordance with another study on several species which showed that primers with (CT), (TC) repeats are more polymorphic than primers with the other di-, tri-, or tetra-nucleotide repeats [15]. However, based on our results, we can confirm the successful usage of tri- and tetra-nucleotides primers as well as the effective usage of primers with (TC) and (CT) di-nucleotide repeats in chrysanthemum.

Microsatellites selected for this investigation were developed previously for *Ch. morifolium* cultivars [4,6] and were used by several researchers for genetic diversity studies. In addition, numerous studies used agarose gels for the visualization of SSR fragments with suitable polymorphism detection [4,6,10]. However, although a multibanding pattern was reported in these SSRs [4,6,10], we have not observed more than four bands in each individual accession. Possibly, the higher annealing temperature used in our work (60 °C) compared with previous studies (54–57 °C) is the reason of the lower band number obtained in our study. Furthermore, we observed 38% of identical DNA-fingerprints using SSR markers, possibly because of the low resolving power of agarose gel electrophoresis which is a limitation of our study. Nevertheless, some authors reported that microsatellite analysis alone can underestimate the genetic diversity because of the homoplasy [23,24]. Thus, combination of the several marker types can help better understanding of the genetic diversity of the species. The total number of bands varied greatly between the markers: 4 (SSR), 15 (ISSR), and 10 (SCoT). More polymorphic fragments were obtained by ISSR markers compared with SCoT and SSR, which is consistent with the other studies [9,20,25]. One of the reasons is the wider range of fragment sizes obtained by ISSR amplification: 300–2100 bp (SCoT) and 250–2800 bp (ISSR). This can also be one of the reasons of better discriminating power of ISSR markers compared with SCoT. As a consequence, the average polymorphism in the collection varied accordingly: 28% (SSR), 69% (ISSR), and 55% (SCoT).

Polymorphism information content (PIC) corresponds to its ability to detect the polymorphism among individuals of a population [26,27]. The PIC maximal value for dominant markers is 0.5 [26,28,29,30,31,32,33,34]; that is what we obtain as output from the online tool [35]. On the other hand, some studies reported a PIC value for dominant markers higher than 0.5 [36,37,38,39]. 

The higher PIC values for the markers ISSR873 and SCoT12 correspond with their equal distribution in the chrysanthemum population and correlate with the higher number of amplified fragments obtained by these markers. Our results indicate that PIC values were higher in SSR markers compared with ISSRs and SCoTs and ranged from 0.4–0.5. However, the efficient PIC for the codominant SSR markers is expected to be 0.5–1.0 [40]. Thus, it can be concluded that all PIC values are too low for SSR markers on chrysanthemum that corresponds with the low total band number, low polymorphism, and low discriminating power of these markers. Discriminating power (D) represents the probability that two randomly chosen individuals have different patterns, and thus are distinguishable from one another [40]. Our results confirmed that the highest discriminating power of the markers corresponds with the higher PIC value and vice versa. 

Diversity index (*h)* is defined as the probability that an individual is heterozygous for the locus in the population and widely used parameter of the marker [41,42]. According to our results, the standard deviations of the mean *h* were too high, indicating that *h* should be assessed separately by each marker rather than by the average of all the markers. In addition, these results confirmed that the genetic diversity parameters are more dependent on the size and content of the analysed dataset rather than the marker type [43,44]. This could be a reason why this parameter has not been correlated with the polymorphism of the marker.

The lowest number of monomorphic bands was observed by ISSR markers (2%) and the highest, by SSR markers (25%). It should be noted that SCoT markers showed 10% of monomorphic bands. For the future studies, it should be considered that P, h, and PIC are very much depended on the sample size. The removal of multilocus identical accessions resulted in different sample sizes for different markers. In this case, marker parameters are not easy to compare with each other.

Interestingly, the contradictory conclusions about the level of polymorphism of SSR, ISSR, and SCoT markers can be met in the different studies. Some studies reported that ISSR markers were more efficient in genetic diversity assessment and produced greater number of polymorphic bands compared with SCoT markers [26,45]. However, in some other studies, greater polymorphism was obtained by SCoT primers compared with ISSR primers in [45,46,47,48]. On the other hand, some researchers reported that polymorphism of SSR and SCoT markers on chrysanthemum was almost similar and a high level of correlations was observed between two marker data in chrysanthemum and they confirmed that the efficiency of SSR and SCoT markers is equal for chrysanthemum [11]. 

### 3.2. Genetic Structure of Germplasm Collection Based on SSR, ISSR, and SCoT Polymorphisms

Following a STRUCTURE HARVESTER analysis, SSR, SCoT, and ISSR markers showed K = 2, K= 6, and K = 3 populations, respectively. The number of K was different, which can be explained by different discriminating power and different genome targets regions amplified by these markers. Among three marker types, ISSR markers showed better genetic structure in our study; however, 20–50% of admixtures were observed in most of the accessions in each of the three clusters. Despite the high percentage of admixture, the cultivars were mostly separated by origin. These results are consistent with other studies which showed that the origin-based separation of genetic clusters was clear in chrysanthemum by ISSR and SSR markers [6,7].

The combined data on three markers did not show a clear genetic structure of the chrysanthemum collection. On the other hand, some studies showed that RAPD, ISSR, and RAPD + ISSR data produced similar results dividing the anise landraces into two main groups and a high level of correlations were observed between different marker data [29]. We speculate that the different ploidy level and a high heterozygosity do not allow to obtain a clear genetic structure in the chrysanthemum collection. Our results are consistent with earlier studies. Li et al. (2016) observed no clear genetic structure related to origin, inflorescence type, or other morphological traits in the chrysanthemum collection based on SCoT markers [9]. Klie et al. (2013) reported the lack of any detectable population structure in *Chrysanthemum morifolium* due to repeated backcrossing [49]. Compared with UPGMA data, many individual clusters were not far away from each other and weak site differentiation among the clusters can be explained due to the low genetic distance among the accessions [46]. Moreover, this indicates a mixed ancestry of chrysanthemum in Russia. Another explanation given for the weak population structure is the effectively maintained geneflow, free pollination technique which was often used in Russian chrysanthemum breeding.

### 3.3. Genetic Diversity and Relationships and Correspondence with Phenotypical Traits 

Compared with STRUCTURE, neighbour joining analysis corresponded better with phenotype based joined dataset of three marker types, showed the best correspondence with phenotype than when analyzed separately by each SSR, ISSR, and SCoT trees. Some branches on the UPGMA-tree combined mostly Holland cultivars, and other branches included the most Russian cultivars and hybrids. Darwin tree comparison showed no correlations between SSR, ISSR, and SCoT trees. This result is consistent with the findings of other researchers [9,47,48,49,50]. A lower level of correlation between these markers could probably reflect that these markers are known to target different genomic regions involving repeat and/or unique sequences, which may have differentially evolved or been preserved during the course of artificial selection.

The morphology of florets and the number of composite flowers in inflorescence are two key traits in the chrysanthemum classification [51]. In our study, the composite flower size and the plant height were discriminative by genetic data; however, no clear separation by the floret type was observed. The results are consistent with Li et al. (2016) who reported that both origin and flower type (disbud vs. spray inflorescence) were distinctive according to the genetic distances among chrysanthemums, however imperfectly [9]. Other authors also showed that flat and tubular florets split in most chrysanthemum genotypes; however, the spoon and abnormal florets mingle with the flat floret type in *C. morifolium* [10]. Recent study suggests that many intermediate floret types exist between the two main flat and tubular types in chrysanthemum. Because these traits are quantitative, there are still many unknown but very important genes controlling these traits that have not yet been discovered [52]. This will limit the breeding of different chrysanthemum flower types. 

For decades, Russian breeding programs have aimed to develop one-head big pompon flower cultivars which were of high demand in the local market. However, in last decade, spray daisy-eyed cultivars have been popular in the composite bouquets; that is why this became another breeding direction. Many cultivars have been developed; however, the hybridization with a pollen mix was the most common breeding technique indicating why the full genetic background of many cultivars is not clear. Our results helped to partly clarify the possible genetic background of some important local cultivars and hybrids. Genetic branch IIIb combined several Russian hybrids (Р-194-13, Р-195-7, Р-195-8, Р-195-9, P-194-12). Р-194-13 and Р-194-12 were derived from ‘Izetka Bernstein’ as maternal genotype, thus it is not surprising that they are in the common branch. Р-195-7, Р-195-8, and Р-195-9 were derived from ♀К-10-3 × ♂ ‘Mona Lisa’. However, their parental genotype ‘Mona Lisa’ occurred in the other branch far distantly. Possibly the pollination with the pollen of ‘Mona Lisa’ was not successful, which is confirmed by the flower phenotype. The cultivar ‘Harlequin’ joined the branch and is often used in local breeding programs on Chrysanthemum due to its valuable traits, e.g., tolerance to biotic and abiotic stresses and reproducibility of the flower traits in the offspring. ‘Noktyurn’ was suspected to be derived from it, which was confirmed by our genetic results. ‘Vesuvio’ with another types of florets also joined the branch IIIb, but the other phenotypical traits (plant height, stem thickness, peduncle thickness, inflorescence, composite flower, floret color, disc color) were similar to its neighbours in the branch IIIb.

Most parts of the other locally derived genotypes occurred in the branch IIIc3 and small spray and anemone-shaped composite flowers are typical for most of the members. The sub-branch IIIc3.1 combined phenotypically similar Holland cultivars ‘Statesman’ and ‘Focus’ (popular ancestors in the Russian breeding programs) with the small anemone-shaped composite flowers. A set of closely related Russian cultivars are possibly derived from ‘Focus’ and have short plant height, small doubled composite flowers, and funnel-shaped florets with yellow-orange color range: ‘Rozovaya Dragocennost’, ‘Nejnost’, ‘Krasnoe Znamya’, and ‘Medeya’. The sub-branch IIIc3.2 combined ‘Sadko’ which is another often used ancestor in Russian breeding programs, and the set of hybrids suspected to be derived from it, namely Р-196-15, Р-196-26, Р-192-12, and Н-103-7. Interestingly, four phenotypically different disbud Holland cultivars such as ‘Mirazh’, ‘Princess Anna’, ‘Rezume’, ‘Etrusko’ and ‘Regina’ joined this group, and possibly were also used as parental genotypes in the open pollination hybridization. 

Based on genetic data, we confirmed that ‘Sadko’, ‘Harlequin’, ‘Vesuvio’, ‘Izetka Bernstein’, and ‘Focus’ were the most often used ancestor genotypes for many locally derived cultivars and hybrids. We suppose that ‘Mona Lisa’, which had been mentioned as one of the parents for many locally derived hybrids, was indeed a misclassified accession. Most likely, ‘Princess Anna’, ‘Rezume’, ‘Etrusko’ and ‘Regina’ were possible parents of some local hybrids.

## 4. Materials and Methods

### 4.1. The Plant Material and DNA Extraction

The plant material of the *Chrysanthemum* × *morifolium* was obtained from the germplasm bank of the FRC SSC RAS (Appendix A). Young and healthy leaves of each accession were collected in 2-mL tubes and dried using silica gel. The leaf material was stored at 4 °C until DNA isolation. The dried leaf material was ground and DNA extraction was performed using CTAB protocol [53]. DNA quality was checked by agarose-gel electrophoresis using Lonza LE2 agarose (Lonza, Basel, Switzerland) and spectrophotometrically using BioDrop µLite (Biodrop, Cambridge, UK)and all samples were diluted to 20 ng µL^−1^ and stored at −20 °C.

### 4.2. Phenotypical Evaluation of Collection

Phenotypical Evaluation of Collection was performed in the period of 2018–2021. The following parameters were evaluated during these years: flowering period, plant height, stem thickness; stem anthocyanidin colour, peduncle thickness; leaf size and shape; leaf colour; leaf base shape; capitulum type, rays shape; rays tip; floret type; floret colour, disk colour; assignment of the cultivar (disbud-type, spray-type, dwarf); inflorescence type, number of composite flowers, and diameter of capitulum and inflorescence. The results were converted into the binary matrix (Appendix A).

### 4.3. Genetic Analysis

Since the 36 SCoT primers were originally developed for *Oryza sativa* and the 10 ISSR primers originally developed for *Camellia sinensis,* we first assessed the transferability of these primers to 7 and 24 *Chrysanthemum* accessions, respectively (Table 3). As soon as the multilocus primers could be transferable to the different plant genera [15,19,20] we evaluated their efficiency for chrysanthemum.

The SCoT PCR reaction mixture consisted of 10 μL 2x HS-TaqPCR reaction buffer (Biolabmix, Novosibirsk, Russia) contained Hot Start Taq-Polymerase, 0.4 μL of primer (10 µM), 2 μL of DNA (20 ng µL^−1^) and DEPC-treated water in a total PCR volume of 20 µL. Amplification was carried out in the MiniAmp thermal cycler (Thermo Fisher Scientific, Massachusetts, U.S) with the following program: primary denaturation 5 min at 95 °C, annealing 35 cycles denaturation at 95 °C for 1 min, aneling at 52 °C for 1 min, elongation at 72 °C for 2 min and the final elongation at 72 °C for 5 min. The separation of SCoT-fragments was performed on a 2% agarose gel for 2.5 h at 90 V in 1 × TAE buffer.

The ISSR PCR reaction mixture consisted of 10 μL 2x HS-TaqPCR reaction buffer (Biolabmix, Novosibirsk, Russia) contained Hot Start Taq-Polymerase, 0.3 μL of primer (10 µM), 1 μL of DNA (20 ng L^−1^) and DEPC-treated water in a total PCR volume of 20 µL. Amplification was carried out in the MiniAmp thermal cycler (Thermo Fisher Scientific, USA) with the following program: primary denaturation 5 min at 95 °C, annealing 40 cycles of 20 sec at 53 °C with elongations at 72 °C for 1 min 45 sec and the final elongation at 72 °C for 7 min. The separation of ISSR-fragments was performed on a 2% agarose gel for 2.5 h at 90 V in 1 × TAE buffer.

Additionally, 5 SSR primer pairs developed for chrysanthemum (Table 3) were used. For SSR analysis, the 20-μL PCR reaction mixture contained 10 μL 2x HS-TaqPCR reaction buffer (Biolabmix, Russia), 0.2 μL of each primer (10 µM), 1 μL of DNA (20 ng µL ^−1^) and DEPC-treated water. Two-step amplification program was used: primary denaturation 5 min at 95 °C, annealing 40 cycles of 15 sec at 50–60 °C and the final elongation at 72 °C for 7 min. The separation of SSR-fragments was performed was performed on a 2% agarose gel for 2.5 h at 90 V in 1 × TAE buffer.

### 4.4. Statistical Analysis

Genetic diversity parameters were calculated for each ISSR and SSR and SCoT locus in the core collection using the software program GeneAlex ver. 6.5 [56,57] and the valuable online resource [35]: the range of the band size for each primer; total number of bands; number of monomorphic bands; P = Polymorphism (%); PIC—polymorphism information content, D—discriminating power; I = Shannon’s Information Index; and genetic diversity (*h*). 

The analysis function ‘Matches’ in GeneAlex ver. 6.5 [56,57] was used to identify genotypes with identical allelic patterns within dataset. Subsequently, the model-based clustering method was applied using the software STRUCTURE ver. 2.3.4. (Oxford, UK) [58]to verify the genetic structure within the chrysanthemum core collection. The parameters were 50,000 burn-in periods and 50,000 Markov Chain Monte Carlo repetitions using the admixture model with correlated allele models. The software program STRUCTURE HARVESTER (California, US) [59] was used for detecting the most likely value for *K* based on Evanno’s ΔK method [60]. Phylogenetic trees were drawn based on the dissimilarity matrix of the genetic and phenotypic (flower traits), respectively, using DARWIN ver.6.0 [61]. Additionally, principal coordinate analysis (PCoA) was performed based on efficient markers for the 95 chrysanthemum accessions in GeneAlex ver. 6.5 with 1000 random permutations. 

## 5. Conclusions

The results of this work showed the high efficiency, good discriminating power, and transferability of several SCoT and ISSR markers for the genetic analysis of *Chrysanthemum morifolium* germplasm. The following SCoT markers were efficient with high polymorphism level: SCoT12, SCoT20, SCoT23, SCoT29, SCoT31, SCoT33, and SCoT34. New efficient ISSRs markers were revealed in chrysanthemum: (GA)8T, (CT)8G, (CTTCA)3, (GGAGA)3, (TC)8C, (CT)8TG. Genetic distances, admixtures, and relationships were established between the Russian and foreign accessions. Several SCoT markers were efficient to separate the groups distinctly according to the phenotypical traits such as plant height (SCoT29, SCoT34), thickness of the stem and peduncle (SCoT31, SCoT34), leaf size, and the petal type (SCoT31). These results are important for the germplasm characterization and for searching markers associated with important traits in chrysanthemum for the trait-oriented breeding.

## Figures and Tables

**Figure 1 plants-10-01302-f001:**
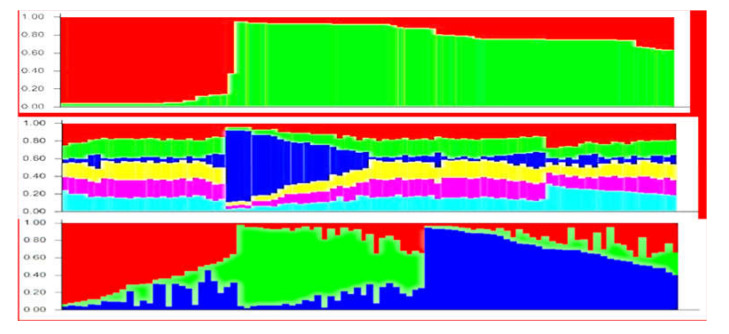
Genetic structure among 95 chrysanthemum accessions based on SSR (**top**) (K = 2), SCoT (**middle**) (K = 6), and ISSR (**bottom**) (K = 3) data, respectively. Each segment represents the estimated membership fraction of each genetic cluster.

**Figure 2 plants-10-01302-f002:**
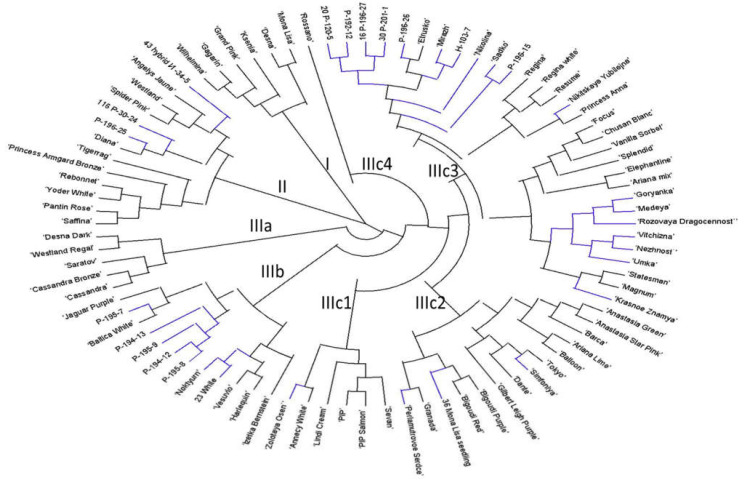
Genetic relationships among 95 chrysanthemum genotypes calculated by SSR, SCoT, and ISSR data. The letters on the picture indicate the related brunches. Blue branches—genotypes of the Russian breeding, black branches—genotypes of the foreign breeding.

**Figure 3 plants-10-01302-f003:**
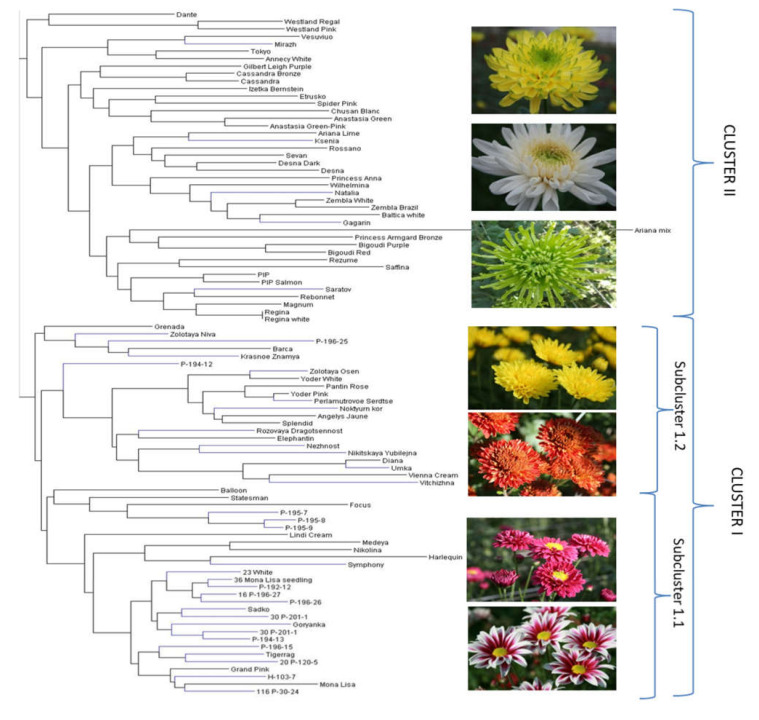
Phenotypic UPGMA–dendrogram of chrysanthemum collection based on 20 important flower traits. Blue branches—genotypes of the Russian breeding, black branches—genotypes of the foreign breeding.

**Figure 4 plants-10-01302-f004:**
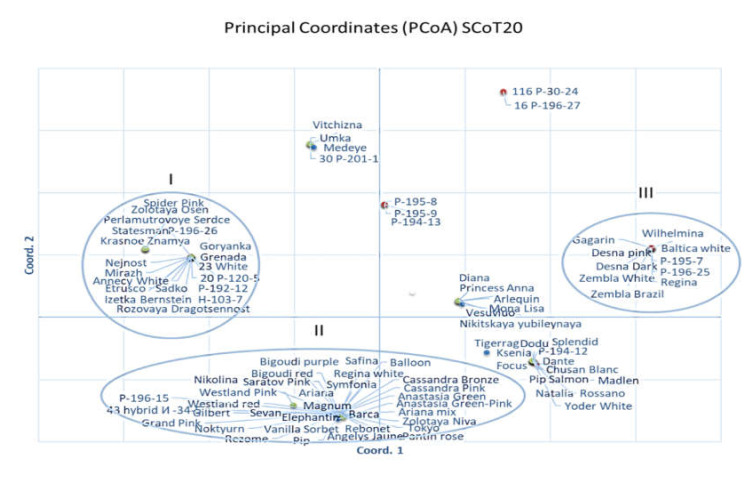
The genetic diversity among 95 chrysanthemum genotypes calculated by SCoT20 (**top**) and SCoT23 (**bottom**) markers. Different numbers indicate the main distant genetic clusters (I, II, III, IV).

**Table 1 plants-10-01302-t001:** Genetic diversity parameters of the selected SSR, ISSR, and SCoT markers in *Ch. morifolium* collection.

Locus	Approx. Band Size, bp	Total Number of Bands	Number of Monomorphic Bands	*P*	*PIC*	*D*	*I*	*h*
**SSR markers**								
gi298296818	1000–1070	4	2	15.6	0.50	0	0.31	0.00
357	250–280	2	1	12.8	0.48	0.27	0.38	0.25
gi298297301	150–160	2	0	14.6	0.47	0.27	0.41	0.25
gi298295865	120–140	4	0	46.5	0.38	0.72	0.63	0.50
320	200–300	4	1	50.3	0.40	0.79	0.33	0.56
MEAN ± SD		3.2 ± 0.1	0.8 ± 0.0	28.2 ± 1.9	0.45 ± 0.05	0.41 ± 0.33	0.41 ± 0.07	0.31 ± 0.22
**ISSR markers**								
ISSR810	250–2000	19	1	48.7	0.33	0.69	0.48	0.49
ISSR813	750–1800	7	1	66.1	0.36	0.90	0.42	0.44
ISSR815	450–1950	15	0	64.8	0.36	0.89	0.44	0.44
ISSR873	300–2800	24	0	92.2	0.44	0.99	0.17	0.15
ISSR880	450–2200	19	0	63.0	0.34	0.85	0.40	0.47
ISSR13	980–1500	5	0	56.5	0.36	0.87	0.52	0.50
ISSR15	500–1600	11	0	77.5	0.40	0.95	0.38	0.51
ISSR814.1	450–1550	18	0	82.3	0.42	0.97	0.31	0.40
MEAN ± SD		14.8 ± 6.6	0.3 ± 0.0	69.0 ± 1.4	0.38 ± 0.04	0.89 ± 0.09	0.39 ± 0.02	0.43 ± 0.12
**SCoT markers**								
SCoT12	300–1750	15	0	73.8	0.42	0.93	0.47	0.39
SCoT20	350–1500	8	2	49.6	0.37	0.75	0.30	0.50
SCoT23	750–1500	4	1	51.6	0.37	0.77	0.43	0.50
SCoT29	500–1850	8	0	45.7	0.38	0.71	0.36	0.51
SCoT31	560–1000	9	2	56.3	0.38	0.81	0.53	0.49
SCoT33	400–1240	14	0	53.4	0.38	0.79	0.47	0.52
SCoT34	880–2100	13	2	51.3	0.38	0.77	0.53	0.51
MEAN ± SD		10.1 ± 4.0	1.0 ± 0.0	54.5 ± 0.9	0.39 ± 0.02	0.79 ± 0.07	0.45 ± 0.03	0.49 ± 0.05

P = Polymorphism (%), PIC – polymorphism information content, D – Discriminating power; I = Shannon’s Information Index; h = Diversity.

**Table 2 plants-10-01302-t002:** Correspondence of the genetic positions with phenotypical traits.

Branch	No of Accessions	Phenotypical Traits Common to More Than 85% of Accessions of the Branch							
		Plant Height	Stem Thickness	Peduncle Thickness	Inflorescence	Composite Flower	Floret Type	Floret Color	Disc Color
I	6	tall	thick, medium	thick, medium	disbud, spray	big, daisy-eyed doubled	strait flat	different	green
II	13	medium tall	thick, medium	medium	disbud, pompon	doubled, medium size	tubular twisted, bounded	pink, red, red-purple	yellow, light yellow
IIIa	5	medium tall	medium	thick	disbud, pompon	big, doubled	bent, hanging tubular	red, red-purple, purple	yellow-green
IIIb	12	different	thin, medium	thin, medium	spray	small, daisy-eyed doubled	Flat	light yellow, yellow, orange	yellow, yellow-orange
IIIc1	6	medium tall	medium	medium	spray, flat corymb	doubled	strait simmetrical flat	different	different
IIIc2	14	medium tall	medium	medium thick	disbud, pompon spray	big, doubled	quilled, broken, incurved	different	different
IIIc3.1	15	short, medium	medium	medium thick	spray, compact corymb	small, doubled, anemone-shaped	funnel-shaped	different	yellow, light yellow, orange
IIIc3.2	16	tall, very tall	medium, thick	medium thick	disbud, spray	daisy-eyed semi-doubled, doubled	Different	pink, red-purple	yellow, yellow-orange
IIIc4	1	very tall	medium	thick	disbud	doubled	Flat	pink	yellow-green

**Table 3 plants-10-01302-t003:** ISSR, SCoT, and SSR primers used for the genetic analysis of the chrysanthemum germplasm collection. The most efficient ones are marked in bold.

Name	Primer Sequence 5′3′	Origin Species	Reference
SCoT Marker			
SCoT1	CAACAATGGCTACCACCA	Oryza sativa	[19]
SCoT2	CAACAATGGCTACCACCC	Oryza sativa	[19]
SCoT3	CAACAATGGCTACCACCG	Oryza sativa	[19]
SCoT4	CAACAATGGCTACCACCT	Oryza sativa	[19]
SCoT5	CAACAATGGCTACCACGA	Oryza sativa	[19]
SCoT6	CAACAATGGCTACCACGC	Oryza sativa	[19]
SCoT7	CAACAATGGCTACCACGG	Oryza sativa	[19]
SCoT8	CAACAATGGCTACCACGT	Oryza sativa	[19]
SCoT9	CAACAATGGCTACCAGCA	Oryza sativa	[19]
SCoT10	CAACAATGGCTACCAGCC	Oryza sativa	[19]
SCoT11	AAGCAATGGCTACCACCA	Oryza sativa	[19]
SCoT12	ACGACATGGCGACCAACG	Oryza sativa	[19]
SCoT13	ACGACATGGCGACCATCG	Oryza sativa	[19]
SCoT14	ACGACATGGCGACCACGC	Oryza sativa	[19]
SCoT15	ACGACATGGCGACCGCGA	Oryza sativa	[19]
SCoT16	ACCATGGCTACCACCGAC	Oryza sativa	[19]
SCoT17	ACCATGGCTACCACCGAG	Oryza sativa	[19]
SCoT18	ACCATGGCTACCACCGCC	Oryza sativa	[19]
SCoT19	ACCATGGCTACCACCGGC	Oryza sativa	[19]
SCoT20	ACCATGGCTACCACCGCG	Oryza sativa	[19]
SCoT21	ACGACATGGCGACCCACA	Oryza sativa	[19]
SCoT22	AACCATGGCTACCACCAC	Oryza sativa	[19]
SCoT23	CACCATGGCTACCACCAG	Oryza sativa	[19]
SCoT24	CACCATGGCTACCACCAT	Oryza sativa	[19]
SCoT25	ACCATGGCTACCACCGGG	Oryza sativa	[19]
SCoT26	ACCATGGCTACCACCGTC	Oryza sativa	[19]
SCoT27	ACCATGGCTACCACCGTG	Oryza sativa	[19]
SCoT28	CCATGGCTACCACCGCCA	Oryza sativa	[19]
SCoT29	CCATGGCTACCACCGGCC	Oryza sativa	[19]
SCoT30	CCATGGCTACCACCGGCG	Oryza sativa	[19]
SCoT31	CCATGGCTACCACCGCCT	Oryza sativa	[19]
SCoT32	CCATGGCTACCACCGCAC	Oryza sativa	[19]
SCoT33	CCATGGCTACCACCGCAG	Oryza sativa	[19]
SCoT34	ACCATGGCTACCACCGCA	Oryza sativa	[19]
SCoT35	CATGGCTACCACCGGCCC	Oryza sativa	[19]
SCoT36	GCAACAATGGCTACCACC	Oryza sativa	[19]
ISSR Marker			
ISSR810	GAGAGAGAGAGAGAGAT	Camellia sinensis	[51]
ISSR813	CTCTCTCTCTCTCTCTT	Camellia sinensis	[54]
ISSR815	CTCTCTCTCTCTCTCTG	Camellia sinensis	[54]
ISSR851	TATTATTATTATTAT	Camellia sinensis	[54]
ISSR873	CTTCACTTCACTTCA	Camellia sinensis	[54]
ISSR880	GGAGAGGAGAGGAGA	Camellia sinensis	[54]
ISSR13	ACACACACACACACACC	Camellia sinensis	[55]
ISSR14	TGTGTGTGTGTGTGTGG	Camellia sinensis	[55]
ISSR15	TCTCTCTCTCTCTCTCC	Camellia sinensis	[55]
ISSR814.1	CTCTCTCTCTCTCTCTTG	Camellia sinensis	[55]
SSR Marker			
gi298295865	F: ACTCACTTGCCCCATTTGTC R: AGAGAAGCTCTCCAGGGACC	Ch. morifolium	[4]
gi298296818	F: ATGTCCAGCTTGATGGGAAG R: GGCCCCTTGCAAATCCTC	Ch. morifolium	[4]
gi298297301	F: TCAAACACCACCACCAACAC R: ATGTCACCAAGTCCTGGTCC	Ch. morifolium	[4]
357	F: ACCCAACCTGAACAAGATGC R: ATACTGCTGCCACTGACCCT	Ch. morifolium	[4]
320	F: GGTCCTTCGTTTCATTTGGA R: CGGGGGTAGGAATAGAAAGC	Ch. morifolium	[4]

## Data Availability

Data is contained within the article and as attached Appendix A.

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
