# Peer review of "Transferability of ISSR, SCoT and SSR Markers for Chrysanthemum × Morifolium Ramat and Genetic Relationships Among Commercial Russian Cultivars"

_plants, 2021, doi:10.3390/plants10071302_

Round 1

Reviewer 1 Report

The manuscript entitled “Transferability of ISSR, SCoT and SSR Markers for Chrysanthemum × Morifolium Ramat and Genetic Relationships Among Commercial Russian Cultivars” by Samarina et al. is interesting and worth publishing. However, the manuscript seems to be novel only at limited features.

Suggestions and corrections

  • In the abstract authors have mentioned 95 cultivars, but in the supplemental table, 96 cultivars were present. The authors should make it clear.
  • The authors should improve the quality of the gel picture. Then authors must include the Marker and Sample numbers in the gel picture?
  • The authors should repeat the experiment and provide good gel pictures with ladders (well resolved).
  • The authors should perform PCR analysis of all the cultivars and show the gel picture.
  • The authors should provide clear images of the dendrogram and PCoA representation.
  • “Table 1” the authors should provide the footnotes of the table information.
  • Line 448- Table No 1 should be changed into Table No 3.
  • The authors should give information about the total number of bands, the number of monomorphic bands, percentage of polymorphism, and polymorphism information content.
  • The authors should comment about ISSR markers were taken from Camellia sinensis and SCoT marker from Oryza sativa.
  • Need more focus on the discussion while comparing the present data with previous studies and refer some recent articles.
  • I would suggest thorough proofreading for rectifying grammatical and usage errors if any.

The manuscript can be considered for publication, only after satisfying all these queries.

Author Response

Dear Reviewer, thank you very much for evaluation of our work, for your useful suggestions and corrections. We appreciate it very much. We have tried to follow the most of your suggestions and revised the article accordingly.

  • In the abstract authors have mentioned 95 cultivars, but in the supplemental table, 96 cultivars were present. The authors should make it clear.

According to this suggestion we have removed one extra cultivar from the Supplementary table 1. This cultivar was not included in the genetic study.

  • The authors should improve the quality of the gel picture. Then authors must include the Marker and Sample numbers in the gel picture? The authors should repeat the experiment and provide good gel pictures with ladders (well resolved). The authors should perform PCR analysis of all the cultivars and show the gel picture.

We have removed the gel picture (Fig1) from the main text and attached new supplementary file with all gel images of three marker types (SCoT, SSR, ISSR) which were selected for the germplasm analysis. In these pictures we added all the numbers of samples (which are corresponds to the list of cultivars in the supplementary table 1). Also we added the size of the markers on the images.

  • The authors should provide clear images of the dendrogram and PCoA representation.

We improved the quality of figures, made two different figures from the Fig3 for better understanding of the position of each cultivar. Also we improved the quality of the Fig 2 and the PCoA figure (now it is Fig 5).

  • “Table 1” the authors should provide the footnotes of the table information.
  • The footnote is provided

  • Line 448 - Table No 1 should be changed into Table No 3.

Done

  • The authors should give information about the total number of bands, the number of monomorphic bands, percentage of polymorphism, and polymorphism information content.

The requested primer parameters are added in the table 1

  • The authors should comment about ISSR markers were taken from Camellia sinensis and SCoT marker from Oryza sativa.

We have added the comment in the Methods section

  • Need more focus on the discussion while comparing the present data with previous studies and refer some recent articles.

We have tried to revise the discussion part according to this suggestion

  • I would suggest thorough proofreading for rectifying grammatical and usage errors if any.

We have carefully check the whole manuscript and revised the grammatical errors

Reviewer 2 Report

The work is very extensive and presents a lot of results. However, it requires some corrections, especially in the part concerning genetic analyzes.
Line 424 - table 3 appears, but table captions do not include such markings. There are two tables labeled Table 1 - this should be corrected.
The authors write about the suitability of selected markers for chrysanthemum research. In my opinion, the work should include several indicators of primers informative. PIC or RP appear in many studies, which allow to determine the polymorphism of primers and their suitability for the assessment of genetic diversity. they also allow the results to be compared with other works. You can also add a table with the number of products obtained for individual primers and the calculated level of polymorphism.
What does Resolving Power Mean in Subheadings? it is not explained in the text.
In table 1, a legend should be added with explanations of individual columns

Author Response

Dear Reviewer, thank you very much for evaluation of our work, for your useful suggestions and corrections. We appreciate it very much. We have tried to follow the most of your suggestions and revised the article accordingly.

  • Line 424 - table 3 appears, but table captions do not include such markings. There are two tables labeled Table 1 - this should be corrected.
    • Revised

  • The authors write about the suitability of selected markers for chrysanthemum research. In my opinion, the work should include several indicators of primers informative. PIC or RP appear in many studies, which allow to determine the polymorphism of primers and their suitability for the assessment of genetic diversity. they also allow the results to be compared with other works.
    • We have changed the table according to this suggestion, we added the PIC, Total bands number, polymorphism. monomorphic bands

  • You can also add a table with the number of products obtained for individual primers and the calculated level of polymorphism.
    • We added this information in the table 1

  • What does Resolving Power Mean in Subheadings? it is not explained in the text.
    • We agree with your note. Resolving power is the ability of the primer to distinguish between the big number of genotypes. Thus we revised the sentence and we use the criteria of Discriminating power which represents the probability that two randomly chosen individuals have different patterns, and thus are distinguishable from one another. We have added this explanation in the footnotes of the revised table 1 as well as in the results description.

  • In table 1, a legend should be added with explanations of individual columns
    • The legend is added

Reviewer 3 Report

This is an interesting piece of work describing the Transferability of ISSR, SCoT and SSR Markers for Chrysanthemum × Morifolium Ramat and Genetic Relationships Among Commercial Russian Cultivars. But, has one conceptual flaw, they compare STRUCTURE and neighbor joining results, under the assumption that they will reflect Chrysanthemums’ phenotypes. This is incorrect because STRUCTURE uses Bayesian inference to shuffle individuals into clusters using genetic data, and NJ is a multivariate classification method based on genetic distances. Neither of them include phenotypic data. So, a priori, there’s no need for branches and clusters to agree, between them or with phenotypes. Probably this is a linguistic misunderstanding that might be corrected clarifying awkward sentences along the manuscript, particularly in the discussion section, and including more details in the statistical analysis section.

Also, even when the authors state in the abstract that “These results provide new findings for the selection of markers associated with important traits in Chrysanthemum for trait-oriented breeding and the implementation of conservation measures.” There is only a line in the conclusion section oriented to conservation, so would be great if the authors could deepen in their conservation recommendations.

Finally, although the general grammar and vocabulary of the text is good, all the text needs minor grammar and vocabulary revision.

Find below a list of suggestions and grammar corrections

L33. add “an” before herbaceous, and “that” before plant.

L37-38. This sentence needs grammar review, please rephrase: Cultivated chrysanthemum is an allo-hexaploid and aneuploid with 37 a most frequent somatic chromosome number of 54 (2n= 6x= 54)

L.42. Replace “tolerant to various stresses” with “, which are tolerant to different types of stress”

L.59-60. Change to “high reproducibility, technical simplicity, and polymorphism”

  1. 64. Replace “presented” with “present”

L.73 Replace “markers based” with “markers are based”

  1. 87 This sentence needs grammar review, please rephrase: These results are an important basis for the implementation of conservation measures and for the trait-oriented breeding of chrysanthemums.
  2. 96 Replace “showed identical amplification patterns with no any polymorphism” with “showed identical but monomorphic amplification patterns”

L.97 Delete “further”

L.112-117 This paragraph is awkwardly written.

  1. 254 Replace “The present study” with “This study”
  2. 267 Replace “differed only at the one nucleotide” with “differed in only one nucleotide”
  3. 269 Replace “with previous” with “with a previous”

L.301, 314 and L. 342 Replace “Comparing” with “Compared”

  1. 327-329 Please rephrase for clarification.
  2. 332. Replace “These results are in consistence with the other studies showed that origin-based separation of genetic clusters” with “These results are consistent with other studies which showed that origin-based separation of genetic clusters”
  3. 342-344 Please rephrase for clarification.

I have also highlighted some grammatical errors in the manuscript.

I hope you find my comments useful.

Author Response

Dear Reviewer,

Thank you very much for your time, for your work with our manuscript. Thank you for your useful suggestions and comments.  As soon as your review came later, than the other two and I did not receive notification, I had a shorter time to work with your comments. However, the first two of your suggestions I will discuss more with the colleagues in our future work. Now, I just can say the following:  

  • This is an interesting piece of work describing the Transferability of ISSR, SCoT and SSR Markers for Chrysanthemum × Morifolium Ramat and Genetic Relationships Among Commercial Russian Cultivars. But, has one conceptual flaw, they compare STRUCTURE and neighbor joining results, under the assumption that they will reflect Chrysanthemums’ phenotypes. This is incorrect because STRUCTURE uses Bayesian inference to shuffle individuals into clusters using genetic data, and NJ is a multivariate classification method based on genetic distances. Neither of them include phenotypic data. So, a priori, there’s no need for branches and clusters to agree, between them or with phenotypes. Probably this is a linguistic misunderstanding that might be corrected clarifying awkward sentences along the manuscript, particularly in the discussion section, and including more details in the statistical analysis section.
    • I’m not sure that I understand correctly this comment. Sure, the algorithms are different, but we wanted to explain why the certain cultivars occurred in the joint genetic cluster. That is why our goal was to find the common phenotypical traits in the cultivars which belongs to a certain genetic cluster. We tried to found the association of the certain marker and the certain amplified fragments of the markers with the phenotypical traits. We discussed these results in 3.3. These results can be useful for the further association analysis between marker and traits that is why we wrote, that the results can be useful for the trait-oriented breeding. The similar analysis can be found in the several studies, which are cited and discussed in the chapter 3.3. Please see, for example:

Li, P.; Zhang, F.; Chen, S.; Jiang, J.;Wang, H.; Su, J.; Fang, W.; Guan, Zh.; Chen, F. Genetic diversity, population structure and association analysis in cut chrysanthemum (Chrysanthemum morifolium Ramat.). Mol Genet Genomics 2016, 291, 1117–1125. https://doi.org/10.1007/s00438-016-1166-3

Luo, C.; Chen, D.; Cheng, X.; Liu, H.; Li, Y.; Huang, C. SSR Analysis of Genetic Relationship and Classification in Chrysanthemum Germplasm Collection. Horticultural Plant Journal 2018, 4(2), 73–82. https://doi.org/10.1016/j.hpj.2018.01.003

  • Also, even when the authors state in the abstract that “These results provide new findings for the selection of markers associated with important traits in Chrysanthemum for trait-oriented breeding and the implementation of conservation measures.” There is only a line in the conclusion section oriented to conservation, so would be great if the authors could deepen in their conservation recommendations. 
    • The study was conducted on the core collection of Chrysanthemum of FRC SSC RAS. It is ex situ collection maintained in the greenhouse and in vitro. We mean that efficient set of markers can help better characterization of the collection and confirmation of the genotypes, that is why we wrote that results will be useful for conservation. Maybe it was no clear so we revised that sentences, for better understanding.

  • add “an” before herbaceous, and “that” before plant.
    • revised
  • L37-38. This sentence needs grammar review, please rephrase: Cultivated chrysanthemum is an allo-hexaploid and aneuploid with 37 a most frequent somatic chromosome number of 54 (2n= 6x= 54)
    • revised
  • 42. Replace “tolerant to various stresses” with “, which are tolerant to different types of stress”
    • revised
  • 59-60. Change to “high reproducibility, technical simplicity, and polymorphism”
    • revised
  • Replace “presented” with “present”
    • revised
  • 73 Replace “markers based” with “markers are based”
    • revised
  • 87 This sentence needs grammar review, please rephrase: These results are an important basis for the implementation of conservation measures and for the trait-oriented breeding of chrysanthemums.
    • revised
  • 96 Replace “showed identical amplification patterns with no any polymorphism” with “showed identical but monomorphic amplification patterns”
    • revised
  • 97 Delete “further”
    • revised
  • 112-117 This paragraph is awkwardly written.
    • revised
  • 254 Replace “The present study” with “This study”
    • revised
  • 267 Replace “differed only at the one nucleotide” with “differed in only one nucleotide”
    • revised
  • 269 Replace “with previous” with “with a previous”
    • revised
  • 301, 314 and L. 342 Replace “Comparing” with “Compared”
    • revised
  • 327-329 Please rephrase for clarification.
    • revised
  • Replace “These results are in consistence with the other studies showed that origin-based separation of genetic clusters” with “These results are consistent with other studies which showed that origin-based separation of genetic clusters”
    • revised
  • 342-344 Please rephrase for clarification.
    • revised
  • I have also highlighted some grammatical errors in the manuscript. I hope you find my comments useful.
    • Thank you very much again for the useful comments. We really appreciate this!

Round 2

Reviewer 1 Report

No further comments.